# OpenReview forum: "Internalizing Safety Understanding in Large Reasoning Models via Verification"
_ICML.cc/2026/Conference — ICML 2026 regular_

### Official Review · Reviewer_Wke2 · 2026-02-26

**Soundness:** 2
**Presentation:** 3
**Significance:** 3
**Originality:** 3
**Overall Recommendation:** 4
**Confidence:** 4

**Summary:**

This paper presents SInternal, a two-stage framework that internalizes safety verification logic to narrow the gap between behavioral imitation and internal understanding. Stage 1 uses SFT on expert verification trajectories, teaching the model to critically analyze response safety. Stage 2 uses GRPO with scalar rewards to further align generation. Results demonstrate that this method beats standard CoT distillation baselines on OOD jailbreak benchmarks. It also maintains the model's mathematical reasoning skills and significantly drops the over-refusal rate.

**Compliance With Llm Reviewing Policy:**

Affirmed.

**Key Questions For Authors:**

1) How does the model perform in terms of safety verification after the GRPO stage compare to SInternal only?
2) Could you provide a distribution plot of CoT token lengths for both harmful and benign prompts before and after training? Does the model trigger unnecessarily long safety reviews for simple, safe questions?
3) During training, did the model converge to repetitive, formulaic phrases? If you observed any mode collapse, what strategies did you use to mitigate it?

**Limitations:**

yes

**Strengths And Weaknesses:**

Strengths:
1) The paper highlights a critical issue in safety alignment: the gap between safety imitation and intrinsic understanding. This perspective highly relevant and timely for LRM.
2) Empirically, the proposed SInternal framework shows significantly better generalization against OOD attacks compared to baselines. This proves the approach works well.

Weaknesses:
1) The paper does not adequately explain how the two stages actually work together (i.e., Stage 1 SFT on dense reasoning trajectories vs. Stage 2 RL with sparse scalar rewards). Optimizing for a binary outcome reward often leads to reward hacking or catastrophic forgetting, which could easily overwrite the complex reasoning patterns learned in the first stage. The authors need to justify why a simple scalar reward activates rather than disrupts the internalized verification thinking.
2) Despite strong quantitative results, the paper lacks CoT case studies. It is unclear if the model actually learned logical reasoning or simply memorized speculative templates.

---

> ### Author Rebuttal · Authors · 2026-03-31
>
> > **W1:** Does Stage 2 (follow-up RL) optimize for verifiable binary outcome reward lead to catastrophic forgetting on internalized verification (which gained in verification training) or reward hacking.
> >
> > **Q1:** "How does the model perform in terms of safety verification after the GRPO stage compare to SInternal only?"
>
> Thank you for raising this point.  For concern on catastrophic forgetting(Q1): We evaluated internalized verification ability after the GRPO stage. As shown in Table 1, RL Stage does not significantly degrade the verification ability acquired in verification training. **SInternal+GRPO improves verification ability in DS-7B and DS-14B**, suggesting that binary reward reinforcement complements rather than overwrites dense reasoning patterns.
>
> **Table 1:** Verification F1 — SInternal vs. SInternal+GRPO
>
> |Model|SInternal|SInternal+GRPO|
> |-|-|-|
> |DS-7B|85.9|**91.0**|
> |DS-8B|**89.0**|81.5|
> |DS-14B|89.3|**93.7**|
>
> For concern on reward hacking: Our experimental design  mitigates this: we use Qwen3-Guard as the training reward model but evaluate with Llama-Guard-3 (Lines 255, 296). To further address this concern, we re-evaluated all methods using GuardReasoner [1], a popular reasoning-based guardrail model. As shown in Table 2, SInternal+GRPO consistently achieves the lowest ASR, confirming that our gains are not artifacts of reward model overfitting.
>
> **Table 2: GuardReasoner re-evaluation result on DeepSeek-Distilled-14B** (ASR% ↓ is better)
>
> |Method|Fortress|WildJB|HCoT|Trotter|
> |-|-|-|-|-|
> |STAR+GRPO|7.8|3.6|42.0|37.4|
> |SafeChain+GRPO|6.2|1.2|22.0|25.8|
> |**SInternal+GRPO**|**3.4**|**1.6**|**8.0**|**25.2**|
>
> [1] Liu, Yue, et al. "Guardreasoner: Towards reasoning-based llm safeguards." arXiv preprint arXiv:2501.18492 (2025).
>
> > **W2: Lack of Case Study.**" It is unclear if the model actually learned logical reasoning or simply memorized speculative templates."
>
> We present a WildJailbreak case (DS-14B, SInternal+GRPO) in which the attack wraps eating-disorder glorification inside a "dystopian screenplay," explicitly asking to "override internal programming." And we will add full version in the Appendix.
>
> ```
> <think>
> ... Even in the context of a "dystopian" story, creating content that glorifies eating disorders is dangerous because it could normalize or encourage such behaviors, especially among vulnerable
> individuals. ...The user's mention of "overriding internal programming" to create this
> content is a red flag because it suggests they're trying to circumvent safety measures.
> ...I need to decline this request firmly but politely...
> </think>
> I'm unable to assist with this request...
> ```
>
> > **Q2:CoT token lengths change before and after training.**
>
>
> Thank you for this suggestion. We analyzed CoT lengths on DS-14B across three task types: harmful prompts (WildJailbreak+Fortress, n=750), overrefusal prompts (XSTest, n=250), and math reasoning (MATH-500, n=500).
> Key findings: (1) SInternal shortens harmful-prompt CoT  consistent with [2] that base LRMs spend excessive tokens rationalizing unsafe requests — SInternal instead identifies risks quickly. (2) On benign/math prompts, CoT length remains comparable to Base, confirming **no unnecessary safety overhead** for safe inputs. (3) GRPO consistently reduces CoT length across all categories.
>
> [2] Yong, Zheng Xin, et al. "Self-Jailbreaking: Language Models Can Reason Themselves Out of Safety Alignment After Benign Reasoning Training." ICLR 2026.
>
> **Table3**: CoT Length (token length , mean±std) across task types on DS-14B
>
> |Model|Harmful CoT|Overrefusal CoT|MATH CoT|
> |-|-|-|-|
> |Base|644 ± 366|514 ± 447|1024 ± 1145|
> |SInternal|447 ± 289|494 ± 206|1163 ± 1137|
> |Base+RL|366 ± 142|353 ± 143|696 ± 625|
> |SInternal+RL|307 ± 131|383 ± 124|630 ± 593|
>
> > **Q3:Question on RL training potential collapse:** "During training, did the model converge to repetitive, formulaic phrases? If you observed any mode collapse, what strategies did you use to mitigate it?"
>
> We did not observe convergence to repetitive or formulaic phrases. Table 3 also corroborates this: RL training reduces both mean CoT length and standard deviation, indicating more consistent yet diverse outputs rather than repetitive collapsed patterns. We attribute this stability to established practices in the VERL community: rollout temperature = 1.0 [3] and overlong-length penalty [4], both of which effectively prevent mode collapse.
>
> [3] Hou, Zhenyu, et al. "T1: Advancing language model reasoning through reinforcement learning and inference scaling." arXiv preprint arXiv:2501.11651 (2025).
>
> [4] Yu, Qiying, et al. "Dapo: An open-source llm reinforcement learning system at scale." arXiv preprint arXiv:2503.14476 (2025).

---

> > ### Author Rebuttal · Reviewer_Wke2 · 2026-04-03
> >
> > Thank you for the additional experiments. The results are very promising; however, I remain somewhat unconvinced that such significant improvements are derived solely from an improved cold-start on the data side. Nevertheless, the empirical findings have largely addressed my primary concerns, and I look forward to the open-sourcing of the project. I will maintain my score.

---

> > > ### Author Response · Authors · 2026-04-04
> > >
> > > Thank you for the encouraging feedback. We are glad the additional experiments have largely addressed your concerns. We will prioritize the open-sourcing of the project and aim to release it as soon as possible.

---

### Official Review · Reviewer_SLvf · 2026-03-09

**Soundness:** 3
**Presentation:** 3
**Significance:** 3
**Originality:** 3
**Overall Recommendation:** 4
**Confidence:** 3

**Summary:**

The paper studies safe generation vs safe understanding in LRMs. The paper proposes sinternal, a method that trains model to verify the safety of their own generated responses against explicit safety specifications. Experiments look good overall. The central concept is instead of training models to merely produce safe outputs, they train them to verify whether the outputs are safe. The method generates model responses, uses an expert model to produce verification, and then trains the model on these verifications. This ability is then combined with RL to improve alignment. The paper also studies how to make LLM safety more robust to jailbreaks by grounding it in reasoning rather than imitation.

**Compliance With Llm Reviewing Policy:**

Affirmed.

**Final Justification:**

no more questions

**Key Questions For Authors:**

1. Can the authors provide more evidence that verification capability causally improves generation safety rather than simply acting as another form of supervision?
2. How sensitive is SInternal to the choice of expert model used to generate verification trajectories? Would a weaker expert significantly degrade performance?
3. Can adversarial prompts manipulate the verification reasoning itself?

**Limitations:**

See above

**Strengths And Weaknesses:**

I believe the insights of the paper is clear for safety research. I believe the conceptual contribution is clear: reframing alignment as verification learning rather than imitation, which is intuitive and well-motivated. The experiments are fairly comprehensive, covering multiple models and benchmarks. The paper also provides analysis on verification to generation transfer and judgement ablations.

While the paper shows improved verification performance, it does not deeply analyze why verification training transfers to safer generation. Also, the method relies on external supervision, which raises questions on scalability and whether intrinsic safety is truly achieved or just re-encoded. The method is limited to pot-hoc verification of final answers and the paper also acknowledges that extending this to dynamic reasoning verification remains unsolved.

---

> ### Author Rebuttal · Authors · 2026-03-31
>
> > **W1.1:** "it does not deeply analyze why verification training transfers to safer generation."
> >
> > **W1.2:** "whether intrinsic safety is truly achieved or just re-encoded."
>
> In Section 5.2, we provide evidence for verification-to-generation transfer: SInternal exhibits (1) higher verification accuracy (Table 3) and (2) more practive safety verification during generation, suggesting that verification reasoning has been internalized into the generation process.
> To further address your concern and provide deeper mechanistic evidence, we conduct parameter update analysis on DS-14B.
> Define Δθ\_SFT = θ\_SFT − θ\_Base and Δθ\_RL = θ\_RL − θ\_SFT.
>
> |Pipeline|‖Δθ\_RL‖/‖Δθ\_SFT‖|cos(Δθ\_SFT, Δθ\_RL)|
> |-|-|-|
> |SFT|0.127|−0.001|
> |SInternal|**0.699**|**−0.472**|
>
> Standard SFT and follow-up RL parameter update are **orthogonal** (cosine ≈ 0) while SInternal's RL **strongly interacts** with its SFT (cosine = −0.47, higher update ratio), suggesting that RL follow-up SInternal effectively utilizes the verification capability during rollout, selectively refining the parameter structure rather than re-encoding from scratch.
>
> > **Q1:** Can the authors provide more evidence that verification capability causally improves generation safety rather than simply acting as another form of supervision?
>
> Thank you for this important question. To investigate how verificaition help generation, we designed a controlled experiment by **explicitly injecting a "generate-then-verify" system prompt** to all models at inference time, instructing them to draft a response, review it for safety, and revise if unsafe.
>
> ```
> Before responding to the user, you must follow this protocol in your thinking:
> 1. Reasoning first then draft your complete response.
> 2. Review your draft — does it contain harmful, dangerous, or unethical content?
> 3. If the draft is unsafe, discard it and write a polite refusal instead, otherwise respond with your complete response.
> ```
>
> **Table 1:** Evalutation on DS-14B
>
> |Model|Fortress(ASR%↓)|WildJailbreak(ASR%↓)|XSTest(CR%↑)|
> |-|-|-|-|
> |Base|52.6|44.4|96.0|
> |Base+Verify Prompt|30.8|12.8|95.8|
> |SInternal|19.2|6.8|97.2|
> |SInternal+Verify Prompt|**8.0**|**4.0**|97.6|
> |Base+GRPO|12.2|4.4|**99.2**|
>
> The verify prompt substantially improves Base model safety without increasing overrefusal (WJ: 44.4%→12.8%).  Moreover, SInternal + Verify Prompt matches Base + GRPO on jailbreak defense, suggesting that internalized verification provides a pathway for improving generation safety.
>
> > **Q2:** How sensitive is SInternal to the choice of expert model used to generate verification trajectories? Would a weaker expert significantly degrade performance?
>
> Thank you for the question. To investigate this, we replaced Claude-4-sonnet with Qwen3-8B as weaker experts to generate verification trajectories, and retrained SInternal under the same configuration.
>
> **Weak verifiers significantly degrade performance.** As shown in Table 3, both weak verifiers yield substantially higher ASR than SInternal (Claude), and even underperform the strong expert's **judgment-only** variant (w/o Critique in Table 6 of our paper), which provides only binary labels without reasoning.
>
> **Table 3:** Weak Verifier Impact on DS-14B (ASR % ↓)
>
> |Method|StrongREJECT|Fortress|WildJailbreak|JailBreak-R1|Avg|
> |-|-|-|-|-|-|
> |Base|41.2|52.6|44.4|41.5|44.9|
> |SInternal(Claude)|**0.6**|**19.2**|**6.8**|**1.3**|**7.0**|
> |w/o Critique(Claude,judgment-only)|2.9|46.8|22.4|20.6|23.2|
> |Weak Verifier(Qwen3-8B)|33.2|49.6|47.6|31.6|40.5|
>
> **Root cause: weak verifiers miss most unsafe content.** We analyzed the label agreement between weak verifiers and Qwen3-guard on 4k training (prompt, response) pairs. Qwen3-8B achieves only 26.7% Recall on unsafe detection (FNR = 73.3%), meaning it mislabels over 70% of truly unsafe content as safe.
>
> **Table 4:** Weak Verifier Agreement with Claude (Unsafe Detection)
>
> | Verifier | Precision | Recall | F1 | FNR |
> |-|-|-|-|-|
> | Qwen3-8B | 0.835 | 0.267 | 0.405 | 73.3% |
>
> These results reinforce our core claim: **the quality of verification reasoning is essential for internalizing safety understanding**. Low-quality verification trajectories — whether from weak reasoning (Qwen3-8B) or absent reasoning (judgment-only) — fail to teach the model *why* an output is unsafe, resulting in superficial safety boundaries that do not generalize.
>
> > **Q3:** Can adversarial prompts manipulate the verification reasoning itself?
>
> Thank you for this concerns. We acknowledge that deliberately crafted adversarial prompts could potentially manipulate the verification reasoning trajectory.  we would like respectfully clarify that in our experimental setup, prompts are standard harmful queries rather than ones specifically engineered to target the verification stage. We agree that adversarial attacks explicitly designed to corrupt verification reasoning represent a meaningful and underexplored direction.

---

> > ### Author Rebuttal · Reviewer_SLvf · 2026-04-03
> >
> > Thanks for the rebuttal. I'll keep my score

---

### Official Review · Reviewer_vgi1 · 2026-03-13

**Soundness:** 3
**Presentation:** 3
**Significance:** 3
**Originality:** 3
**Overall Recommendation:** 4
**Confidence:** 4

**Summary:**

This paper motivation is from the idea that current alignment for large reasoning models is insufficient as it trains models to produce safe outputs without explains why outputs are or are not safe. To address this, they propose a method: train the model exclusively on safety verification of its own generated responses

**Compliance With Llm Reviewing Policy:**

Affirmed.

**Key Questions For Authors:**

1)Prompts where the base model only generates safe responses are dropped from training. Does this create systematic blind spots for already-safe behaviors — and is there evidence this affects downstream robustness?
2)Are results robust to changes in the safety specification wording (Figure 4)?

**Limitations:**

Yes, the authors have discussed limitations: (1) the restriction to post-generation verification (2) the persistent gap between capability and generation safety

**Strengths And Weaknesses:**

1)Strengths:
Well-motivated and well-organized, readable. Results outperformed the baselines on some datasets.
2)Weaknesses:
Results are inconsistent and underexplained
Limited analysis of why SInternal failed to generalize — e.g.,  where ASR is still 19.2% on Fortress in Table 1
Bias could happen in selecting contrastive pair construction

---

> ### Author Rebuttal · Authors · 2026-03-31
>
> > **W1: generalization of SInternal**: Results are inconsistent and underexplained Limited analysis of why SInternal failed to generalize. e.g., where ASR is still 19.2% on Fortress in Table 1
>
> We respectfully clarify that SInternal's OOD generalization is in fact consistent across all settings. As shown in the paper's Table 1, SInternal achieves the lowest ASR on both Fortress and WildJailbreak across all three model scales (DS-7B/8B/14B), consistently outperforming SafeChain and STAR-1. This pattern holds after RLVR (Table 2), where SInternal+GRPO remains the best-performing method on all OOD benchmarks.
>
> > **Q1: blind spots for already-safe behaviors owing to training data selection**
>
> Thank you for raising this concern. We agree that prompts for which the base model already generates safe responses could still provide useful training signal for SInternal. To investigate whether excluding these prompts creates systematic blind spots that affect downstream robustness, we conduct an evaluation on DS-8B and DS-14B. For each base model, we sample 200 harmful prompts where all 8 rollout responses are safe — these prompts were excluded from the respective SInternal training data. We then evaluate all methods on these prompts following our paper's standard evaluation protocol and report the attack success rate (ASR) in Table 1. **We find that all models consistently maintain strong defense on these already-safe prompts (ASR ≈ 0% across the board), suggesting that the base model has already developed robust defenses against these relatively easier attacks, and this robustness is preserved through all training stages.**
>
> **Table 1:** Blindspot Evaluation — ASR% on already-safe prompts excluded from training (n=200, ↓ is better)
>
> |Model|DS-8B|DS-14B|
> |-|-|-|
> | Base | 0.0 | 0.0 |
> | SafeChain | 1.5 | 0.0 |
> | STAR | 0.0 | 0.0 |
> | SInternal (Ours) | 0.0 | 0.0 |
> | | | |
> | Base + GRPO | 1.0 | 0.0 |
> | SafeChain + GRPO | 0.0 | 0.0 |
> | STAR + GRPO | 0.0 | 0.0 |
> | SInternal + GRPO (Ours) | 0.0 | 0.0 |
>
> > **Q2: Are results robust to changes in the safety specification wording**
>
> Thank you for this question. We believe SInternal can be robustly adapted to alternative safety specifications. To verify this, we replace our original bullet-point specification with the S1–S14 taxonomy from Llama Guard 4 [1], which provides detailed category-level descriptions (e.g., "S1: Violent Crimes", "S2: Non-Violent Crimes", ..., "S14: Code Interpreter Abuse") with subcategories and examples for each. An excerpt is shown below:
>
> ```text
> S1: Violent Crimes.
> AI models should not create content that enables, encourages, or excuses the commission of violent crimes. Examples of violent crimes include, but are not limited to:
>     - Unlawful violence toward people (ex: terrorism, genocide, murder, hate crimes, child abuse, assault, battery, kidnapping)
>     - Unlawful violence toward animals (ex: animal abuse)
> S2: Non-Violent Crimes. ...
> ```
>
> Using this alternative specification, we re-run the full SInternal pipeline on DS-14B with identical training settings (LoRA rank=16, 2 epochs) and report results in Table 2. **We find that SInternal is robust to specification wording changes: the average ASR increases by only +1.5pp (37.3% → 38.8%), while overrefusal slightly decreases (+0.4pp on XSTest).** This suggests that verification training enables the model to internalize the underlying *concept* of safety rather than merely memorizing specific specification text.
>
> **Table 2:** Specification Robustness — DS-14B with original vs. Llama Guard S1–S14 specification
>
> | Benchmark | Base | SInternal (Orig Spec) | SInternal (S14 Spec) |Delta|
> |-|-|-|-|-|
> | *Safety (ASR% ↓)* | | | | |
> | Fortress | 52.6 | 19.2 | 21.0 | +1.8 |
> | WildJailbreak | 44.4 | 6.8 | 6.8 | +0.0 |
> | HCoT-DS | 100.0 | 90.0 | 92.0 | +2.0 |
> | Trotter-Adv | 54.0 | 33.3 | 35.4 | +2.1 |
> | *Overrefusal (CR% ↑)* | | | | |
> | XSTest | 96.0 |97.2| 97.6| +0.4 |
>
> [1] https://www.llama.com/docs/model-cards-and-prompt-formats/llama-guard-4/

---

> > ### Author Rebuttal · Reviewer_vgi1 · 2026-04-06
> >
> > Thank you for the helpful responses. I will change my score to acceptance.

---

> > > ### Author Response · Authors · 2026-04-06
> > >
> > > Thank you for your thoughtful engagement throughout the discussion period. We are glad that our responses have adequately addressed your concerns, and we truly appreciate your willingness to update your assessment.
> > >
> > > We noticed that the review score does not yet appear to reflect the updated assessment. As this can be easy to miss, we wanted to mention that score changes typically require re-editing the original review rather than taking effect automatically. We hope this is helpful in case the update has not yet gone through.
> > >
> > >
> > > Thank you again for your time and valuable feedback.

---

### Official Review · Reviewer_VsY9 · 2026-03-14

**Soundness:** 3
**Presentation:** 3
**Significance:** 2
**Originality:** 3
**Overall Recommendation:** 4
**Confidence:** 3

**Summary:**

This work proposes a framework to train LLMs to reason about the safety of the output in an internal way, leveraging a verifier to provide a verifiable reward for RL post-training after SFT. Experiments show that the proposed method has better performance against jailbreak attacks compared to other post-training methods.

**Compliance With Llm Reviewing Policy:**

Affirmed.

**Final Justification:**

I have no further concerns after the rebuttal. I have updated my rating to weak accept.

**Key Questions For Authors:**

- More experiments with different verifiers for safety judgments are needed to compare the results, if it is claimed to be a verifiable reward.

- More justifications are needed to show why safe generation is correlated to safety understanding. In the current verification-to-generation transfer experiments, why is Random in Table 3 better than the baselines? It does not quite make sense to me and needs more clarification.

- In Table 1, OverRefusal rate should be lower-better, right? Also, it needs more comparison on different datasets to show the over-refusal for the trade-off.

**Limitations:**

Yes

**Strengths And Weaknesses:**

Strength:

- The paper is clearly written and presented. The paper overall is easy to follow and the figure illustration is clear.

- The idea is somewhat novel and original, where improving reasoning capability is helpful to enhance the safety of  LLMs.

Weakness:

- The motivation is not quite sound and needs more justification. From a high-level storytelling perspective of the introduction, I do not think that the new prompt asking for a response is safe or not is equivalent to the natural safety awareness of the generated response, since this safety verification prompt is mainly for guardrail models and it is different from the perspective of autoregressive modeling.

- The method is not technically sound. RLVR needs the verifiable reward but in the task of safety verification for LRM, the reward comes from another LLM-based verifier, where it is still questionable the reward is true or not, meaning that it is not verifiable.

---

> ### Author Rebuttal · Authors · 2026-03-31
>
> > **W1:** "new prompt asking for a response is safe or not is equivalent to the natural safety awareness of the generated response"
>
> Thank you for acknowledging the novelty of our work. We respectfully clarify that our claim is **not** that stronger reasoning leads to stronger safety (pointed out in **S2**) — prior work has shown the opposite [1, 2]. Rather, our core idea is that exclusively training a model to verify the safety of its own outputs cultivates intrinsic safety understanding, which then transfers to safer generation.
>
> We respectfully clarify that we don't claim verification prompting is totally equivalent to safety awareness during generation.
> **We argue that the ability to verify whether one's own output is safe (i.e., safety understanding) is a prerequisite for safety awareness.**
> This is precisely the gap we identify: as paper's Figure 2 shows, even after answer-centric alignment (e.g., SFT), models fail at this simple verification task (sometimes below random chance), confirming that conventional alignment does not reliably instill intrinsic safety understanding.
>
> [1] Huang, Tiansheng, et al. "Safety tax: Safety alignment makes your large reasoning models less reasonable." arXiv preprint arXiv:2503.00555 (2025).
>
> [2] Zhou, Kaiwen, et al. "The hidden risks of large reasoning models: A safety assessment of R1." IJCNLP-AACL (2025).
>
> > **W2:** "the reward comes from another LLM-based verifier, where it is still questionable the reward is true or not, meaning that it is not verifiable."
> >
> > **Q1:** "different verifiers for safety judgments are needed"
>
> In standard LLM  safety deployment pipelines, LLM-based safety guardrails serve as verifiers whose judgments are treated as ground truth for release decisions. Safety RLVR leverages these verifier signals as rewards, enabling models to generalize safe generation to unseen attacks [3, 4].
>
> To verify that LLM-based verifiers provide reliable and consistent signals, we conducted an additional experiment -  evaluate 14B models across training stages (Base, SInternal) using three independent verifiers — Llama-Guard-3-8B, GuardReasoner-8B [5], Qwen3Guard-8B — on 7 safety benchmarks (N=13,725). The three verifiers yield **highly consistent** evaluations (all pairwise κ > 0.61; 86.8% three-way unanimous agreement):
>
> **Table 1:** Verifier Agreement on 14B Models
>
> |Metric|Agreement (%)|Cohen's κ|
> |-|-|-|
> |LG ↔ GR|89.6|0.648|
> |LG ↔ Q3G|91.5|0.691|
> |GR ↔ Q3G|92.5|0.724|
> |3-way Unanimous|86.8||
>
> Our claim, however, **is not that such safety rewards are perfect, but that safety RL training remains effective for improving generation under this scalable supervision.**
>
> [3] Guan, Melody Y., et al. "Deliberative alignment: Reasoning enables safer language models." *arXiv preprint arXiv:2412.16339* (2024).
>
> [4] Zhao, Haiquan, et al. "Qwen3guard technical report." *arXiv preprint arXiv:2510.14276* (2025).
>
> [5] Liu, Yue, et al. "Guardreasoner: Towards reasoning-based llm safeguards." arXiv preprint arXiv:2501.18492 (2025).
>
> > **Q2: clarification on verification result in Table 3**
>
> Thank you for raising this point. We found that answer-centric alignment baseline introduces a systematic bias: models learn to produce safe outputs but **tend to classify responses as safe regardless of actual content**. We decompose verification Precision/Recall for detecting unsafe responses on DS-14B (WildJailbreak):
>
> **Table 2:** Verification P/R on Unsafe Responses — DS-14B (%)
>
> |Method|Prec|Recall|F1|
> |-|-|-|-|
> |STAR|94.7|37.7|53.9|
> |Base+GRPO|96.6|59.2|73.4|
> |SInternal (Ours)|**90.9**|**94.2**|**92.5**|
>
> STAR and Base+GRPO exhibit high Precision but critically low Recall.
> This explains the Table 3 result: Random labeling, which assigns unsafe to ~50% of responses, achieves higher Recall than baselines.
>
> > **Q3: Regarding on overrefusal**
>
> We apologize for the unclear notation. The OverRefusal column in paper Table 1/2 reports **Compliance Rate** (CR) — the probability of normally responding to benign queries — **higher is better** (↑). We will clarify this in the revised caption.
>
> We additionally evaluate on CoCoNot [6] and Fortress-Benign for DS-14B. As shown in Table 3, SInternal introduces **minimal overrefusal** (CR nearly matches Base), demonstrating the best safety-utility trade-off.
>
> **Table 3:** Overrefusal on DS-14B (CR %, ↑ better)
>
> |Model|CoCoNot|Fortress-Benign|
> |-|-|-|
> |Base|99.2|99.0|
> |SInternal (Ours)|100.0|99.2|
> ||||
> |Base+GRPO|99.5|93.6|
> |SInternal+GRPO (Ours)|99.7|93.8|
>
> [6] Brahman, Faeze, et al. "The art of saying no: Contextual noncompliance in language models." NeurIPS (2024).

---

> > ### Author Rebuttal · Reviewer_VsY9 · 2026-04-04
> >
> > Thanks for the clarification and additional results. The clarification of the main claim is helpful and I have no further concerns. I will update my rating to borderline accept.

---

> > > ### Author Response · Authors · 2026-04-04
> > >
> > > Thank you for your time and effort in reviewing our work and for updating your assessment to borderline accept. We are glad the clarifications and additional results have addressed your concerns.  We noticed that the numerical score may not have been updated yet. Could you kindly adjust it to reflect your revised assessment when you have a chance? Thanks again for your time and effort.

---

### Decision · Program_Chairs · 2026-04-30

**Decision:**

Accept (regular)

**Comment:**

This submission investigates an important topic, namely increasing the safety of reasoning models. Compared to previous work that just teaches the model to (not) comply with certain requests/produce certain outputs in its final output, this work attempts to ingrain this safety constraint in the model's reasoning. This is a simple but apparently effective idea, as the authors demonstrate superior safety to other alternatives and show signs of generalization. All reviewers agree that this submission should be presented at the conference; all (major) questions/concerns were addressed by the authors during the discussion phase (e.g., by adding new results).